# A Comparison of the Loading Direction for Bending Strength with Different Wood Measurement Surfaces Using Near-Infrared Spectroscopy

**Yohei Kurata**

College of Bioresource Sciences, Nihon University, 1866 Kameino, Fujisawa, Kanagawa 252-0880, Japan; kurata.youhei@nihon-u.ac.jp; Tel.: +81-466-84-3668

**Abstract:** Wood is widely used throughout society for building resources and paper. To further expand wood's use in the wood industry, we tested the bending strength properties of wood and certified its internal quality by using near-infrared spectroscopy (NIRS). In this study, the relationship between bending strength and loading direction was compared by changing the light acquisition point of wood surfaces to elucidate the anisotropy of the wood using NIRS. The two loading directions were defined by using a bending test as the radial section and the tangential section. Two light acquisition points with NIRS were also defined by a bending test as the loading position (the compression surface) and the opposite surface (the tensile surface), and a comparison was made between the prediction accuracy of the wood's mechanical strength properties obtained via a bending test using two pieces of light acquisition data. The strength properties of the wood bending tests were the elastic modulus in bending ($E_b$), the bending strength ($F_b$) and density (DEN). *Cryptomeria japoni*ca was prepared and cut into a final size of 20 mm × 20 mm × 320 mm. Near-infrared (NIR) spectra were obtained from the compression force side and the tensile force side (calculating these averages), and a partial-least-squares regression (PLSR) was performed for the regression analysis. In the NIR measurement position, the best calibration results of the PLSR were the averaged data between the side undergoing the compression force and that undergoing the tensile force. Comparing the two loading directions, the result for the radial section was slightly superior to that of the tangential section. The radial section showed a good relationship between the spectra acquisition position and the arrangement of the wood's structure. The estimation accuracy of bending strength properties differed depending on the location where the NIR spectra acquisition was performed.

**Keywords:** near-infrared spectroscopy; elastic modulus in bending; bending strength; *Cryptomeria japoni*ca; wood anisotropy; partial least squares regression

## 1. Introduction

Near infrared spectroscopy (NIRS) is widely used for nondestructive measurement with agricultural commodities, including wood. The use of wood is essential to our lives because wood was not only a building material, but also a stock of paper. Therefore, wood properties such as bending strength, compressive strength and chemical composition are important for using the right wood in the right place. It had been shown that many wood properties can be measured non-destructively using near infrared spectroscopy; and industrial applications are being attempted.

Both wood mechanical properties as the bending test and chemical components like cellulose, hemicellulose and lignin are simultaneously measured by using NIRS [1–4]. The measurement principle of NIRS spectroscopy is partly explained by assignment the wavelengths in the near

infrared (NIR) region of the spectrum and chemical constituents of wood [5]. Furthermore, in assuming the practical use in the wood industry, near-infrared spectrum is measured from the wood sliding on the belt conveyor and the wood bending strength properties are estimated from such spectral information [6]. Recently, the relationship between NIR measurement methods and wood mechanical properties—including wood anisotropy—have been summarized in detail [7].

In this study, wood anisotropy under loading with bending test was evaluated when the optical measurement position was changed. Two surfaces (the loading position and the opposite side) were measured via NIRS. In general, as the loading position on the wood's surface in the bending test experiences a compression force, the opposite side experiences a tensile force. The used load side and opposite-side spectroscopic data and the averaged load and opposite-side spectroscopic data are compared to the predictions of the wood bending strength properties from the NIRS calibration model. To further explore regarding wood anisotropy, two types of wood samples were prepared—radial section and tangential sections. The loading patterns for the radial section and the tangential section and the two wood surfaces were studied. Prediction accuracy when using NIRS may differ according to the measurement position due to differences in wood-cell accumulation patterns and the wood's anisotropy. Thus, NIRS measurement data for the loading position and the opposite side may affect the prediction accuracy.

## 2. Materials and Methods

### 2.1. NIR Device and Wood Samples

The NIR measurement device and the position of light source and detector (sample holder) are shown in Figure 1. The measurable range of the NIR device (S-7100; Soma Optics, Ltd., Tokyo, Japan) was from 1200 nm to 2500 nm. The resolution of the spectroscope was 1 nm. As the samples were irradiated with monochromatic light, the reflectance spectra were obtained. The number of scans was 5.

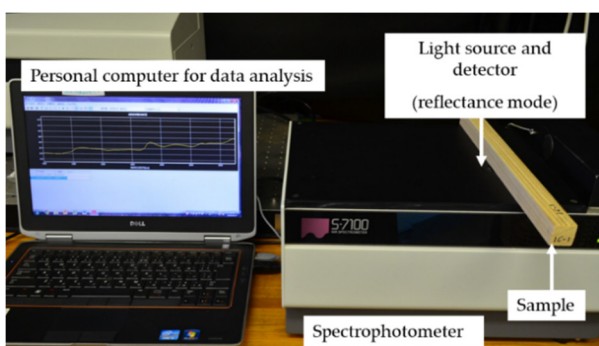

**Figure 1.** Outline of the near-infrared (NIR) measurement device.

*Cryptomeria japoni*ca *D. Don* (obtained from a commercial Japanese lumber mill; located at Fujisawa City, Kanagawa Pref.) was prepared and cut into a final size of 20 mm × 20 mm × 320 mm after wood drying. Its size was suitable for the Japanese industrial standards (JIS) standard with the bending test (Figure 2a). The central concentrated load method was used for bending test (see Figure 2b). The total number of analyzed wood samples was 140. In the log-bucking method, the rift-cutting wood sample was selected as frequently as possible. The sample's length was oriented in the longitudinal direction, and the measurements were performed at room temperature.

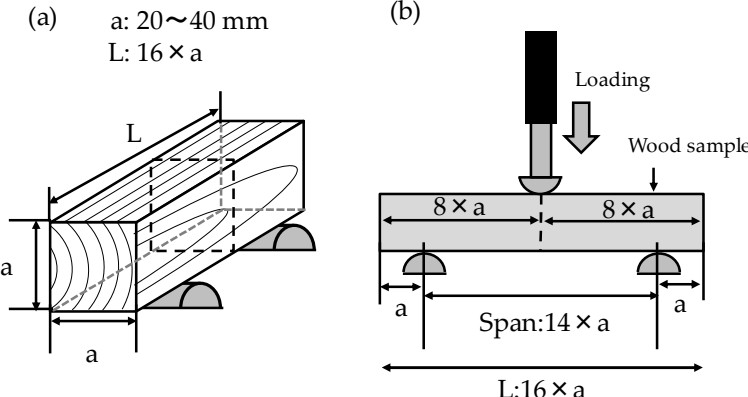

**Figure 2.** Sample size (**a**) and the outline of the bending test (**b**) for using small wood specimen with Japanese industrial standard test.

### 2.2. Measurement of the Wood's Bending Strength Properties Obtained from Bending Test

Bending tests were performed after NIR measurements. The direction of the wood samples during the bending test was divided into two types. The bending test method included a central concentrated load test. The shear force was generated at the center of the wood, but this method was also used because the test sample was small. The elastic modulus in bending ($E_b$), bending strength ($F_b$) and density (DEN) were measured to ensure the wood's bending strength properties obey the JIS [8,9]. To evaluate the $E_b$, the incremental deflection $\Delta y$ for an incremental load $\Delta p$ was selected from the linear elastic area. $E_b$ was obtained using the following formula:

$$E_b = \frac{\Delta p l^3}{4 \Delta y b h^3}, \tag{1}$$

where $l$ (mm) is the span, $b$ (mm) is the width and $h$ (mm) is the thickness of the rectangular sample. The bending strength was calculated as follows:

$$F_b = \frac{3 P_{max} l}{2 b h^2}, \tag{2}$$

where $P_{max}$ (N) is the value of the applied load at failure. These values were obtained by using a precision universal testing machine (AG-X plus 50 kN; Shimadzu, Tokyo, Japan). Both Equations are widely used in the wood science field to calculate bending strength. Figure 3a shows displays the setup for the universal machine, where a compression test, tensile test, bending test, etc., can be performed by changing the metal jig. The loading force of the wood sample and the wood deformation due to load were recorded until the wood failed. Figure 3b illustrates both the load and force directions inside the wood as a result of the bending test. Two forces occurred on the wood's surface due to the bending load. The loading surface on the wood was defined as the compression force side. The opposite side was defined as the tensile side. The load speed was 14.7 N/mm² per minute. $R = 75$ mm steel stock was used as the load point. The DEN was estimated by the division of sample's weight and volume, which was measured by multiplying length and cross-sectional area.

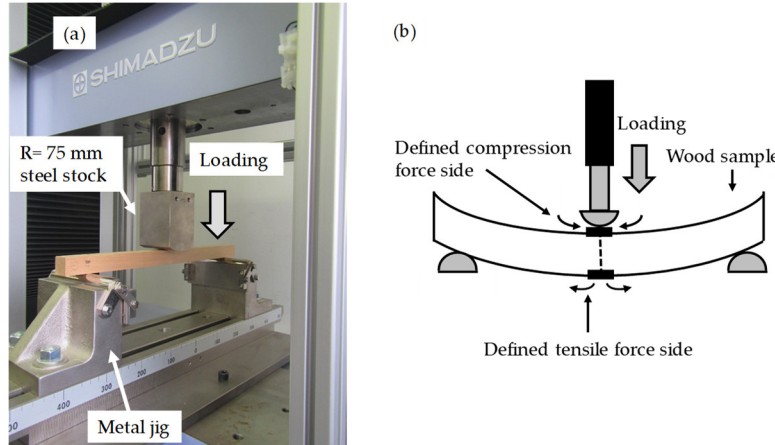

**Figure 3.** Outline of the wood bending test. (**a**) Photograph of the universal testing machine and (**b**) relationship between the loading direction and the force inside the wood.

### 2.3. NIR Spectra Acquisition and the Defined Wood Direction in the Bending Test

In total, 140 wood samples were divided into two groups according to the wood's direction. Seventy samples were radial samples, and another 70 samples were tangential samples. The NIR measurement position and light detection area are shown in Figure 4a. The two sides of the wood were measured by NIRS for each sample in the radial and tangential wood directions. Figure 4b illustrates the relationship between the load direction and the direction of the wood's annual rings. The wood surfaces that took the load were defined as the radial sample and the tangential sample according to the wood's cutting direction from loading. The loading direction for the tangential section test was from the bark side (see Figure 4b). In NIR light acquisition and bending test, the wood sample was measured in an air-dried condition. The temperature was ranged 22–25 °C, and the relative humidity was 40%–55%.

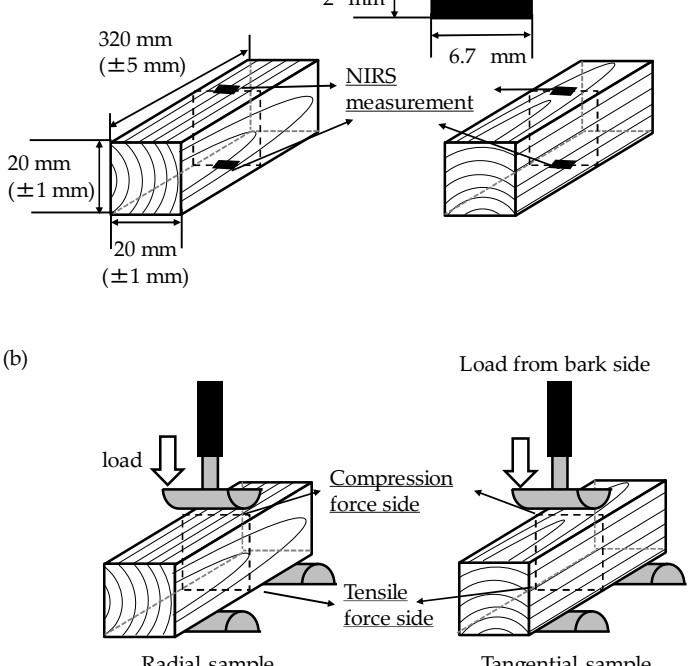

**Figure 4.** Illustration of (**a**) near-infrared spectroscopy (NIRS) measurement position and (**b**) loading direction of the wood sample.

*2.4. Multivariable Analysis Using PLSR for the NIR Data Set*

Statistics of wood bending strength properties for the calibration set and prediction set are summarized in Table 1.

**Table 1.** Outline of wood sample numbers for the calibration set and the prediction set for each measurement condition.

| Compression Surface | | Calibration Set | | | | Prediction Set | | | | |
|---|---|---|---|---|---|---|---|---|---|---|
| | | n | Min. | Max. | Ave. | S.D. | n | Min. | Max. | Ave. | S.D. |
| **Radial section (R)** | $E_b$ (GPa) | 49 | 8.68 | 23.3 | 14.8 | 3.67 | 20 | 8.77 | 22.7 | 15.1 | 4.66 |
| | $F_b$ (MPa) | | 38.4 | 106.9 | 66.6 | 19.2 | | 46.19 | 106.1 | 71.0 | 23.6 |
| | DEN (g cm⁻³) | | 0.328 | 0.638 | 0.431 | 0.0731 | | 0.347 | 0.602 | 0.444 | 0.0927 |
| **Tangential section (T)** | $E_b$ (GPa) | 50 | 6.46 | 23.2 | 15.7 | 4.56 | 20 | 6.92 | 20.9 | 14.9 | 4.26 |
| | $F_b$ (MPa) | | 41.3 | 105 | 72.8 | 20.1 | | 46.2 | 98.4 | 69.2 | 17.7 |
| | DEN (g cm⁻³) | | 0.338 | 0.575 | 0.455 | 0.0782 | | 0.341 | 0.559 | 0.446 | 0.0734 |

n—number of samples; Min.—minimum; Max.—maximum; Ave.—average; S.D.—standard deviation; DEN—wood density.

In the radial sample, one sample was mis-tested and eliminated. The calibration set was used to build the regression model, and the prediction set was used to compare the prediction accuracy. The calibration set included the high and low values of each parameter value. The average and standard deviation values were almost the same between the calibration set and the prediction set for each treatment. The raw NIR spectra often exhibited a baseline shift or drift due to variations in measurement. Other factors could also affect the spectra, such as instrument stability, temperature, humidity or the surface conditions of the sample [10]. Therefore, data pretreatment was an important step in the data analysis to eliminate such errors. Some basic pretreatments included moving average smoothing, multiplicative scatter correction (MSC), and using the second derivative of data. Pretreated NIR spectra (moving average smoothing with a segment size of 13; MSC; a second derivative with a segment size of 13 and mean centering) were used for this analysis [11,12]. Many pretreatments for NIR spectra have been proposed, but only the above pretreatment was used to focus on the force propagation on the wood. Moving average smoothing eliminated the random noise in the NIR spectra. MSC compensated for both the multiplicative effect and the additive effect of the spectra. The second derivative also reduced the multiplicative effect and the additive effect of the spectra and amplified the peaks buried in the spectra. Mean centering involved subtracting the mean from a variable to eliminate the signal's random noise.

Partial least squares regression (PLSR) was performed as the regression analysis. The final number of factors used for each calibration was recommended by the Unscrambler X software (version 10.1; Camo Software, Oslo, Norway) [13]. For the PLSR, the spectroscopic data of the compression side, the tensile side and the average of the compression side and the tensile side were used. The prediction accuracy was compared by using spectroscopic data for the radial and tangential directions. The data were evaluated based on the correlation coefficient ($R^2$ for the calibration set and $R_p^2$ for the prediction set) between the predicted and measured values. In addition, the root mean square error of cross validation (RMSECV) and the root mean square of prediction (RMSEP) were statistically evaluated and compared for calibrations and predictions [1,6,14,15]. The ratio of performance to deviation (RPD), calculated as the ratio of the standard deviation of the reference data to the standard error of prediction, was used to determine the predictive ability of the calibrations. Determination of the RPD allows one to compare the calibrations developed for different wood properties that have different data ranges and units; the higher the RPD, the more accurately the data were fitted by the calibration.

## 3. Results and Discussion

In the calibration dataset of Table 1, the $E_b$, $F_b$ and DEN values did not change with the loading direction (i.e., radial or tangential). Figure 5 shows the raw NIR spectra (a) and the pretreated spectra (b) of the *Cryptomeria japonica* from the radial section and the tangential section from the compression side and the tensile side. The NIR spectra obtained from the radial and tangential wood sections had different absorbance baselines, as shown in Figure 5a. To calculate the pretreatment shown in Figure 5b, as the difference of the absorbance baseline was eliminated, more signal peaks were detected beyond the raw NIR spectra. Therefore, pretreatment allowed observation of additional spectral information.

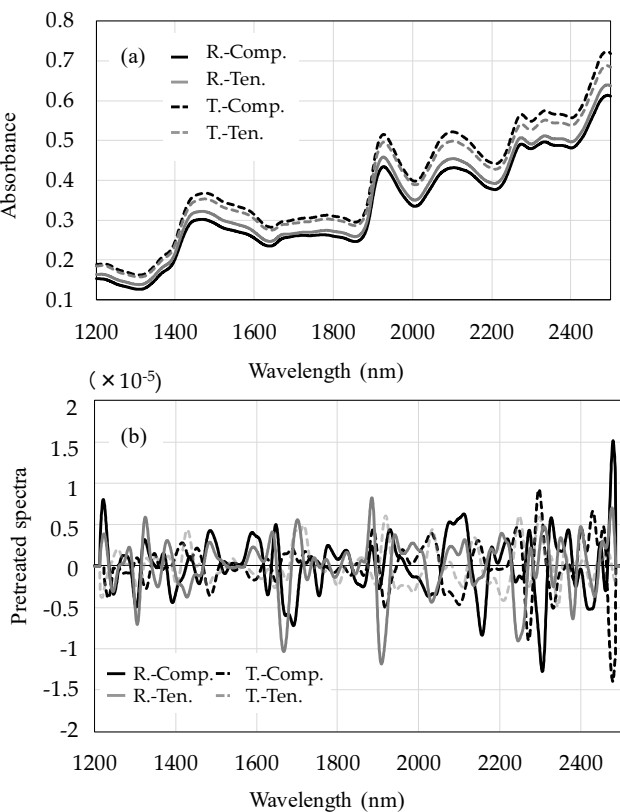

**Figure 5.** Near-infrared spectra of *Cryptomeria japonica* from the radial section; the tensile section from the compression side and the tensile side, (**a**) the raw spectra and (**b**) the pretreated spectra. R—radial sample; T—tangential sample; Comp.—compression surface; Ten.—tensile surface.

Table 2 shows the PLSR results for each of the mechanical strength properties with loading in the radial direction using the pretreated NIR spectra. In the calibration results of the PLSR for the compression surface of the radial section, the $R^2$ values ranged from 0.784 to 0.920, and the RMSECV ranged from 0.0370 to 6.85. In the prediction results, the $R_p^2$ values ranged from 0.710 to 0.852, and the RMSEP ranged from 0.0420 to 9.34. The RMSEP was influenced by the standard deviations of the objective variable. As the standard deviations of the wood's strength parameters were different, the RPD values were calculated to take these differences into account when comparing to the PLS results. The RPD ranged from 1.97 to 3.07.

**Table 2.** Partial least squares regression (PLSR) results for wood's mechanical strength properties using the pretreated NIR spectra with loading of the radial section.

| Compression Surface | Spectra Measurement | Property | Factors | Calibration Set | | Prediction Set | | |
|---|---|---|---|---|---|---|---|---|
| | | | | $R^2$ | RMSECV | $R_p^2$ | RMSEP | RPD |
| Radial section | Compression force side | $E_b$ (GPa) | 6 | 0.870 | 1.42 | 0.820 | 1.74 | 2.68 |
| | | $F_b$ (MPa) | 6 | 0.900 | 6.50 | 0.822 | 8.90 | 2.65 |
| | | DEN (g cm⁻³) | 6 | 0.770 | 0.0400 | 0.710 | 0.0470 | 1.97 |
| | Tensile force side | $E_b$ (GPa) | 7 | 0.846 | 1.53 | 0.776 | 1.90 | 2.45 |
| | | $F_b$ (MPa) | 7 | 0.890 | 6.85 | 0.790 | 9.34 | 2.53 |
| | | DEN (g cm⁻³) | 7 | 0.784 | 0.0390 | 0.722 | 0.0450 | 2.06 |
| | Averaged compression and tensile side | $E_b$ (GPa) | 6 | 0.900 | 1.25 | 0.852 | 1.52 | 3.07 |
| | | $F_b$ (MPa) | 6 | 0.920 | 5.74 | 0.844 | 7.94 | 2.97 |
| | | DEN (g cm⁻³) | 6 | 0.800 | 0.0370 | 0.750 | 0.0420 | 2.21 |

$R^2$—coefficient of determination for calibration set; RMSECV—root mean square error of cross validation; $R_p^2$—coefficient of determination for prediction set; RMSEP—root mean square of prediction; RPD—ratio of performance to deviation.

Figure 6 shows the relationships between the measured value and the NIR-predicted values for each physical property resulting from the pretreated spectra when loading the radial section. As the predictions of the compression data were superior to those of the tensile side data, some sample numbers were near the target line of the calibration curve.

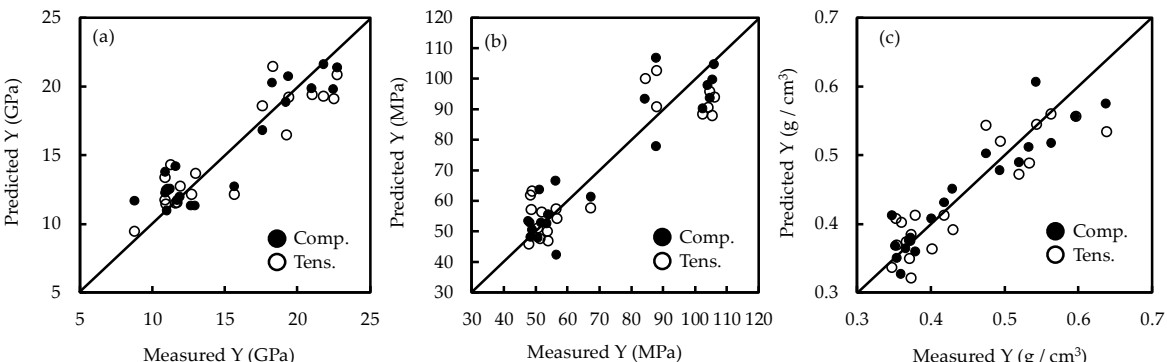

**Figure 6.** Measured versus NIR-predicted value plots for (**a**) modulus of elasticity in bending ($E_b$); (**b**) bending strength ($F_b$) and (**c**) wood density (DEN) for pretreatment NIR spectra of *Cryptomeria japonica* with loading of the radial section. Closed black circles indicate the compression side (Comp.) and the open circles indicate the tensile force side (Tens.).

Table 3 presents the PLSR results for loading of the tangential section using the pretreated NIR spectra. The $R^2$ values ranged from 0.729 to 0.887 and the RMSECV ranged from 0.0360 to 7.24. In the prediction results for the tensile force side of the tangential section, the $R_p^2$ values ranged from 0.630 to 0.822 and the RMSEP ranged from 0.0442 to 9.49. The RPD ranges between 1.56 and 2.28.

**Table 3.** PLSR results for the wood's mechanical strength properties using the pretreated NIR spectra with loading of the tangential section.

| Compression Surface | Spectra Measurement | | Calibration Set | | | Prediction Set | | |
|---|---|---|---|---|---|---|---|---|
| | | | Factors | $R^2$ | RMSECV | $R_p^2$ | RMSEP | RPD |
| Tangential Section | Compression force side | $E_b$ (GPa) | 5 | 0.863 | 1.64 | 0.802 | 1.97 | 2.16 |
| | | $F_b$ (MPa) | 5 | 0.858 | 7.24 | 0.815 | 8.34 | 2.12 |
| | | DEN (kg m$^{-3}$) | 5 | 0.765 | 0.0360 | 0.678 | 0.0442 | 1.66 |
| | Tensile force side | $E_b$ (GPa) | 5 | 0.841 | 1.76 | 0.786 | 2.05 | 2.08 |
| | | $F_b$ (MPa) | 5 | 0.868 | 6.97 | 0.754 | 9.49 | 1.87 |
| | | DEN (kg m$^{-3}$) | 5 | 0.738 | 0.0388 | 0.630 | 0.0472 | 1.56 |
| | Averaged compression and tensile side | $E_b$ (GPa) | 5 | 0.869 | 1.60 | 0.822 | 1.87 | 2.28 |
| | | $F_b$ (MPa) | 5 | 0.887 | 6.44 | 0.811 | 8.41 | 2.10 |
| | | DEN (kg m$^{-3}$) | 5 | 0.729 | 0.0394 | 0.647 | 0.0448 | 1.64 |

$R^2$—coefficient of determination for calibration set; RMSECV—root mean square error of cross validation; $R_p^2$—coefficient of determination for prediction set; RMSEP—root mean square of prediction; RPD—ratio of performance to deviation.

Figure 7 shows the relationships between the measured value and the NIR-predicted values for each of the physical properties resulting from the pretreated spectra when loading the tangential section. While the compression data appear near the target line, the tensile data are slightly off the target line. To date, the accuracy of wood density prediction has been worse than other mechanical property values. This result tendency was consistent with the result of other researchers [6,11].

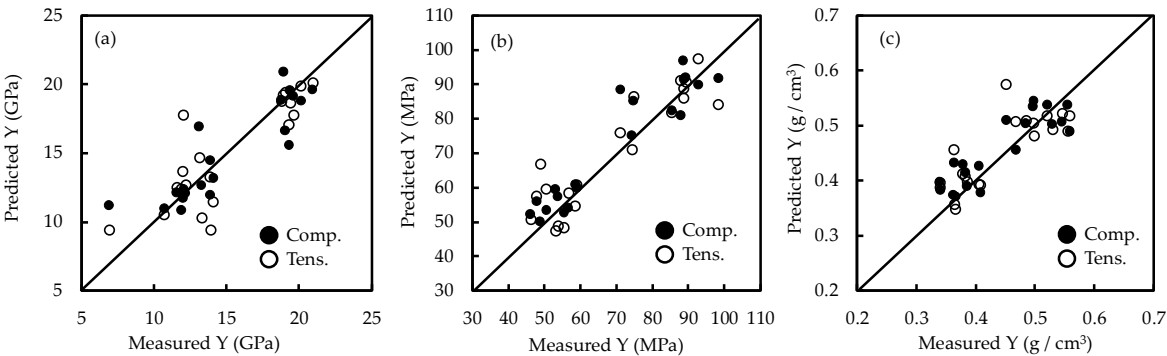

**Figure 7.** Measured versus NIR-predicted value plots for (**a**) $E_b$, (**b**) $F_b$ and (**c**) wood density (DEN) from the pretreated NIR spectra data obtained from the tangential section. The closed black circles indicate the compression side (Comp.) and the open circles indicate the tensile force side (Tens.).

Next, since the number of factors was the same, the tangential section data were focused on. The regression coefficient was calculated to investigate the relationship between the wavelength and each spectroscopic measurement position. Figure 8 shows the spectral plot of the regression coefficients for the PLS models that predict each physical properties of the tangential section. The black line indicates the compression force data, the gray line indicates the tensile force data, and the light gray indicates the averaged data. In Figure 8, although the peak value was either high or low, the trend of the regression coefficient for the compression data and the tensile data were the same.

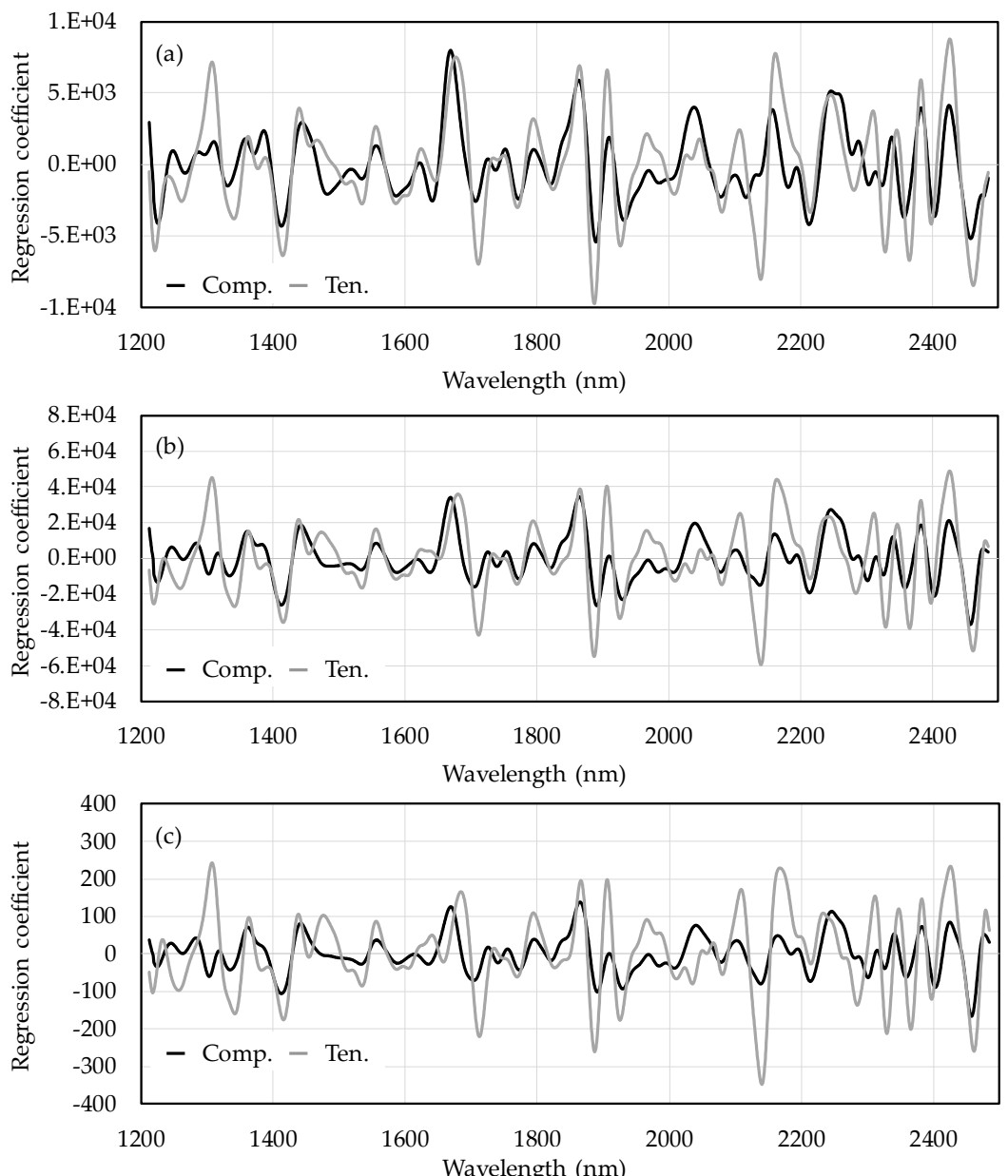

**Figure 8.** Regression coefficients for the PLSR predicting $E_b$ (**a**); $F_b$ (**b**) and DEN (**c**) calculated from the pretreated NIR spectra. Black line indicates the compression force (Comp.) and the gray line indicates the tensile force side (Ten.).

Focusing on the two loading directions, the radial section result was slightly superior to the tangential section result. The radial section showed a good relationship between the spectra acquisition position and the arrangement of the wood's structure. Because the tangential section had a laminated structure—with different densities of early wood and late wood—it is possible that diffuse reflection light was acquired only on the surface layer.

Comparing the spectra measurements for the three cases (the compression side, tensile side and average) of the two loading types, the averaged data obtained the best results by using the $R_p$ and RPD values. Furthermore, the compression side result was better than the tensile side result. Comparing the spectra measurement positions between the compression side and the tensile side, the location where the force was applied on the tensile side varied depending on the direction of the wood fibers. Therefore, the breaking position of the tensile side did not always coincide with the back side of the loading position. In other words, the position at which spectra were acquired and the break position did not correspond. It is presumed that the prediction accuracy on the tensile side was

lower than that on the compression side. Accuracy may have been improved by calculating the average value because the calculation of the average value removed the random noise caused by the stray light and resulted in a stable signal. As the prediction accuracy of the averaged data were better than that of the other two measurement sides, the PLSR results could be improved by changing the measurement method, such as the two measurement positions and averaged data used in this study.

## 4. Conclusions

In determining the relationship between the NIR measurement position and mechanical strength properties of wood with a bending test, this article clarified that changing the position at which spectra are measured can alter the prediction accuracy. A wood sample (*Cryptomeria japoni*ca) obtained from a commercial Japanese lumber mill was used. The physical properties during a bending test were estimated by using the measurement data for a few positions under near-infrared spectroscopy. In the results, the calculation of the average value improved the PLSR prediction accuracy, and the compression side result was slightly better than that for the tensile side. However, no significant difference could be confirmed in the prediction accuracy between each property and the spectra acquisition position. When considering the application of measuring the physical properties of wood via NIRS under practical settings in a wood factory, the results showed that it was not necessary to consider the cross-section and orientation of the cut wood. In addition, only the wood density had a lower prediction accuracy than the other bending strength properties. This study indicates that the location on the wood at which the measurement is performed is important. It is presumed that the wood density prediction accuracy in this study was not high because the small sample size offered few measurement points.

**Funding:** This research was funded by Nihon University's individual research funding.

**Conflicts of Interest:** The author declares no conflict of interest.

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
