# Peer review of "A Comparison of the Loading Direction for Bending Strength with Different Wood Measurement Surfaces Using Near-Infrared Spectroscopy"

_forests, doi:10.3390/f11060644_

Round 1

Reviewer 1 Report

Near infrared spectroscopy (NIR) is a technique widely used for the prediction of different chemical-physical features of wood.

In this article the relationship between bending strength and loading direction was compared by changing the light acquisition point of wood surfaces to elucidate the anisotropy of the wood using NIRS.

The paper is well-structured and has clear purpose and certain conclusions, which have considerable statistics of experimental data. There are several comments:

1. The various comparison of approaches are made in the paper. But there is absolutely no comparison of the data obtained in the works of other authors. This generate questions, since a sufficient number of works related to the topic are known. Moreover, the author cites some of them in the overview.

2. In my opinion, the review does not sufficiently cover world achievements and can be expanded.

3. Line 89-92. Theory and formulas belong to the author of the paper or the developed theory and approaches have been used? Please clarify this point.

I would like to thank the author for an interesting and original paper, which makes a significant contribution to physical properties analysis of wood.

Author Response

At the beginning I would like to thank for revision my article. I hope that I gave you sufficient answers (below).

  1. The various comparison of approaches are made in the paper. But there is absolutely no comparison of the data obtained in the works of other authors. This generate questions, since a sufficient number of works related to the topic are known. Moreover, the author cites some of them in the overview.

>As the reviewer pointed out, there were few data comparisons and there were many subjective expressions. I've included a sentence here to compare and insert with other people's data.

Line205-206: So far, the accuracy of wood density prediction has been worse than other mechanical property values. This result tendency was consistent with the result of reference [6,11].

Although there is no novelty in the experimental method and approaches, I think that it is important to make a case-specific measurement in order to consider the anisotropy of wood. The cited paper does not measure cases separately.

  1. In my opinion, the review does not sufficiently cover world achievements and can be expanded.

>I added some information and improve the sentence in the Introduction section(rewrote the beginning of introduction) and Materials and method section(added newly Figure 2). I think that researchers who measure the wood samples using near infrared spectroscopy will be interested in this paper. Especially the result of changing the load direction is interesting. Since this paper is an open access journal, I would like to leave the level and impact of the paper to the world reader.

  1. Line 89-92. Theory and formulas belong to the author of the paper or the developed theory and approaches have been used? Please clarify this point.

> The theory and formulas are generally known in wood science and therefore not the original of the author. I added this information.

Line97-98: The formula Eb and Fb was commonly used in the wood science.

Reviewer 2 Report

  • Line 41 – quite dated references. There are newer studies you can include. Examples: Estimating the Basic Density and Mechanical Properties of Elite Loblolly Pine Families with Near Infrared Spectroscopy (Acquah et al. 2018), Mechanical properties of wood materials using near-infrared spectroscopy based on correlation local embedding and partial least-squares (Yu et al 2019), High Throughput Screening of Elite Loblolly Pine Families for Chemical and Bioenergy Traits with Near Infrared Spectroscopy (Acquah et al. 2018), Comparison of Methods for Estimating Mechanical Properties of Wood by NIR Spectroscopy (Schimleck et al 2018).
  • Line 71 – Using an average of 5 scans (at a position) is pretty low. In the literature, researchers typically average 16 scans, with others using as many as 64 scans. Do you have a reference for why 5 scans was adopted?
  • Line 75 – Provide more information about wood material. Which region were trees harvested from OR where is the lumber mill located? How old were trees? How was drying done? What was the moisture content of wood samples at time of bending tests?
  • Give full name of JIS as it is mentioned for the first time here.
  • Provide a summary of the dimensions, parameters and conditions that are specified in the “Japanese Industrial Standards Methods of Test for Woods Z:2101” for bending tests. Some of your readers might not have ready access to this standard for reference.
  • The entire length of a test sample is 320mm. What was the span for the bending test?
  • What criteria was used to split the samples into calibration and prediction sets?
  • RPD is used to evaluate how well a calibration will perform during a ‘future’ prediction. So you should include RPDs for the calibration set as well.
  • Line 233 - Wood is anisotropic as you say and it is expected to act different on the different faces, so yes, position/location of spectra collection will affect the results. Wood scientists know this already that is why when they use whole wood specimen in NIR/mechanical studies, they collect spectra from several positions (not just 2 spots as was done in this study) on all the surfaces which they then average into 1 spectrum for that test sample for further analysis. Others have even gone further by milling the whole specimen after mechanical testing and then collecting spectra of the powdered wood to compensate for wood anisotropy and heterogeneity. So I don’t agree it is your study that will ‘clarify’ this already known property of wood.

Author Response

At the beginning I would like to thank for revision my article. I really appreciate you gave me so many comments. It is extremely important to me to make this paper better and have opportunity to publication in journal such as Forests. I hope that I gave you sufficient answers (below).

ï‚·  Line 41 – quite dated references. There are newer studies you can include. Examples: Estimating the Basic Density and Mechanical Properties of Elite Loblolly Pine Families with Near Infrared Spectroscopy (Acquah et al. 2018), Mechanical properties of wood materials using near-infrared spectroscopy based on correlation local embedding and partial least-squares (Yu et al 2019), High Throughput Screening of Elite Loblolly Pine Families for Chemical and Bioenergy Traits with Near Infrared Spectroscopy (Acquah et al. 2018), Comparison of Methods for Estimating Mechanical Properties of Wood by NIR Spectroscopy (Schimleck et al 2018).

> I added new references[reference 3,4] . Thank you for your information.

ï‚·  Line 71 – Using an average of 5 scans (at a position) is pretty low. In the literature, researchers typically average 16 scans, with others using as many as 64 scans. Do you have a reference for why 5 scans was adopted?

> I understand that the recent increase in the number of scans is a trend in spectroscopic measurement. Certainly, I think about 16 times is appropriate as your suggestion. I would like to increase the number of scans and measure next time. I changed some sentence.

Line71-72:Priority was given to the experimental speed, and the number of scans was set to 5

ï‚·  Line 75 – Provide more information about wood material. Which region were trees harvested from OR where is the lumber mill located? How old were trees? How was drying done? What was the moisture content of wood samples at time of bending tests?

> As you pointed out, I think the basic information is important in performing wood research. However, I could not obtain information about the place of origin or the age information because of the use of commercial wood sample. I think it's a good way to know the origin of the wood by purchasing a log of raw wood, processing the wood, drying and obtaining a sample. I would like to get a wood sample in such a way if the budget allows. I could not do that in this study.

As it came to drying wood, the drying method and temperature were important. The steam drying method was often used for commercial wood in Japan, but the temperature was unknown.

The water content of the wood during the bending test was not measured. However, it was estimated to be around 15% because of air dried wood condition. You wondered if a few percent difference in moisture content affected wood mechanical property. PLSR statistical processing decreased the error, so please forgive my answer.

Line 118-119: In NIR light acquisition and bending test, the wood sample was measured in an air-dried condition.

ï‚·  Give full name of JIS as it is mentioned for the first time here.

I am very sorry I made a rudimentary mistake. I should have stated at the beginning of the sentence. I inserted the beginning part.

Line77: Its size was suitable for the Japanese Industrial Standards (JIS) standard with the bending test (Figure 2(a)).

ï‚·  Provide a summary of the dimensions, parameters and conditions that are specified in the “Japanese Industrial Standards Methods of Test for Woods Z:2101” for bending tests. Some of your readers might not have ready access to this standard for reference.

>Since there is no satisfactory web site for JIS in English, I added a new Figure 2 of the bending test method based on the standard. The test sample size and the test information could be shown in Figure 2(Line 82).

ï‚·  The entire length of a test sample is 320mm. What was the span for the bending test?

>There was no information of the street span of your pointed out. I added the span information in Figure 2(b) (Line 82)..

ï‚·  What criteria was used to split the samples into calibration and prediction sets?

>The maximum and minimum values of the objective variable were made into calibration set, and the others were randomly divided into calibration set and prediction set. After division, I calculated the standard deviation of the explanatory variables of each set. If the standard deviation of the objective variable was too different for calibration and prediction, I re-selected the sample. However, there was no clear standard for standard deviation variation in calibration and prediction.

ï‚·  RPD is used to evaluate how well a calibration will perform during a ‘future’ prediction. So you should include RPDs for the calibration set as well.

>Thank you for your instruction. I reconfirmed how to calculate RPD. It was calculated by dividing the square root of the prediction standard error of the prediction set by the square root of the standard deviation of the objective variable. As you pointed out, what kind of formula should I use to calculate the RPD with the values of the calibration set? In addition, is the name of RPD good in that case? I did not know about this.

ï‚·  Line 233 - Wood is anisotropic as you say and it is expected to act different on the different faces, so yes, position/location of spectra collection will affect the results. Wood scientists know this already that is why when they use whole wood specimen in NIR/mechanical studies, they collect spectra from several positions (not just 2 spots as was done in this study) on all the surfaces which they then average into 1 spectrum for that test sample for further analysis. Others have even gone further by milling the whole specimen after mechanical testing and then collecting spectra of the powdered wood to compensate for wood anisotropy and heterogeneity. So I don’t agree it is your study that will ‘clarify’ this already known property of wood.

>When measuring powdered wood, the wood anisotropy would disappear. Since wood powder was used when measuring the chemical composition of wood, created a calibration from the absorption spectrum of wood powder. Therefore, the measurement of the wood powder state was effective in chemical analysis.

I thought that the condition of a wooden board varied from after felling to becoming a wooden board. I also thought that there were many different methods of near infrared spectroscopy depending on what time point was measured and which objective variable was to be predicted. As you pointed out, it was not a clarification, it was just a devision of measurement. I changed the expression from ‘clarify’ to just ‘improved’(Line 235,238).

Actually, I thought that the force propagation in the wood would change by changing the loading surface during the bending test, and it could be captured by near infrared spectroscopy. However, there was few differences in the experiment of changing the loading surface, but I thought this experiment had some significance.

Reviewer 3 Report

Dear author, several improvements need to be made. 

Please, see below my comments. 

INTRO:

L35-38: Awkward paragraph. It needs improvement. I recommend breaking up an introductory paragraph, then starting talking about
NIRS. The way it is written now is like a maze.

L39: Which advantages?

L40: ... producing a great deal of knowledge. This is not scientific writing.

The authors really need to work on this intro. It is poorly written.

L40-41: Is that your objective already, or is it previous research, I think it is the latter. But, it's confusing.

L43: "Extensively discusse". It means several studies have previosuly been done. the authors cited only one.

OVERALL: The authors do need to work on this paper to get it published. Organize your thoughts. Make outlines.

MATERIALS AND METHODS:

I advice to start writing regarding materials first, then methods.

L68: When the author's say "location", it is vague. Why 1200 to 2400 interval was chosen?

L75-76: How about density? Defects? final moisture content? Heartwood or sapwood? Number of growth rings?

L76: What does JIS mean?

L80: To my knowledge, bending IS NOT a physical property. This is a scientific mistake.

L82-85: Draw a diagram to make this readable, then you can use this description as the figure's caption.

L101: was Density calculated as volume?

L108: isn't that grain direction?

L119: Move the table right after you call it.

L136: does Partial least squares regression have a model??? How did you determine significance?

L141-146: Describe any formulation used.

Table1: Did the author consider to make a balanced division between the compression surface? What I mean by that is. Did the
treatments have a common variable (let's say weight) such that radial section and tangential section had the minimal variation possible?

OVERALL: There are several important aspects that the authors did not mention.

RESULTS AND DISCUSSION:

I think the authors did a great job in separating the tables. The authors well presented the results. However, a better discussion
needs to be done. Explain the why of things is important for a complete understaing. This was somewhat done from L213 TO L230.
But, it not enough.

OVERALL: It is good, but there is room for improvement.

CONCLUSIONS

We have good points in the midst of the conclusions that can be fully developed as a better sentence. Working more on that.
Good points are: "When considering the application...". "This study indicates..."

Author Response

Dear author, several improvements need to be made. Please, see below my comments.

>Thank you for the detailed review. I got much your suggestions and ideas for future research. Thank you again. I hope that I gave you sufficient answers (below).

INTRO:

I would like to apologize for the misunderstanding because the text in the introduction part became difficult to understand. The introduction part was completely rewritten to tell the meaning easily to the reader as much as possible.

L35-38: Awkward paragraph. It needs improvement. I recommend breaking up an introductory paragraph, then starting talking about NIRS. The way it is written now is like a maze.

>As you pointed out, the text was difficult to understand. I started writing from near infrared spectroscopy.

L39: Which advantages?

>It was obscure expression. I changed this part.

L40: ... producing a great deal of knowledge. This is not scientific writing.

>I deleted this section.

The authors really need to work on this intro. It is poorly written.

>I changed this part to obeying your suggestion.

L40-41: Is that your objective already, or is it previous research, I think it is the latter. But, it's confusing.

>It was an ambiguous expression.

L43: "Extensively discusse". It means several studies have previosuly been done. the authors cited only one.

> It was as you pointed out. Only the final version, which had undergone various examinations and discussions, was listed here.

OVERALL: The authors do need to work on this paper to get it published. Organize your thoughts. Make outlines.

The introduction has been reorganized and rewritten. I made new introduction from the beginning of the near-infrared spectroscopy so that I could read it and understand its meaning.

Line34-41: Near infrared spectroscopy (NIRS) was widely used for nondestructive measurement with agricultural commodities including wood. The use of wood was essential to our lives because wood was not only a building material, but also a stock of paper. Therefore, wood properties such as bending strength, compressive strength, and chemical composition were important for using the right wood in the right place. It had been shown that many wood properties could be measured non-destructively using near infrared spectroscopy and industrial application was being attempted.

Both wood mechanical properties as the bending test and chemical components like cellulose, hemicellulose and lignin were simultaneously measured by using NIRS [1-4].

MATERIALS AND METHODS:

I advice to start writing regarding materials first, then methods.

> I changed some sentences as much as possible, but couldn't completely rewrite as your instruction

L68: When the author's say "location", it is vague. Why 1200 to 2400 interval was chosen?

>I changed this section. “location” indicated sample holder and light source.

Line68-69: The NIR measurement device and the position of light source and detector (sample holder) are shown in Figure 1.

The definition of near infrared light was from 800 nm to 2500 nm. Since the measurable range of the device was from 1200 nm to 2500 nm, I used those wavelength.

Line69-70: The measurable range of the NIR device (S-7100; Soma Optics Ltd., Tokyo, Japan) was from 1200 nm to 2500 nm.

L75-76: How about density? Defects? final moisture content? Heartwood or sapwood? Number of growth rings?

Density was described at Table 1. Final moisture content was not measured, but the wood was the air dry condition. Heartwood and sapwood were in a mixed state, mostly sapwood. Number of growth rings was not measured. I think the items pointed out to you are important elements. Some of the items were not measured this time, so the description was omitted.

Line 118-119:In NIR light acquisition and bending test, the wood sample was measured in an air-dried condition.

L76: What does JIS mean?

>I added some information. Thank you for your instruction.

Newly Figure 2 was added in Line 82 to describe the JIS standard.

L80: To my knowledge, bending IS NOT a physical property. This is a scientific mistake.

Physical properties have been removed for misleading terms. “The wood strength properties obtained from bending test” or “wood bending strength property” and so on were used in this paper. Thank you for instruction.

L82-85: Draw a diagram to make this readable, then you can use this description as the figure's caption.

New Figure 2(b) was inserted to instruct the bending test(Line 82).

L101: was Density calculated as volume?

>Yes, I added this information.

Line106-107: The DEN was estimated by the weight, length, and dimensions of the sample in air-dried conditon.

L108: isn't that grain direction?

>Yes. I think the reader will understand if the reader combines it with the following Figure 4 (Line121).

L119: Move the table right after you call it.

>Thank you for your instruction. I changed the location of Table 1(Line 126).

L136: does Partial least squares regression have a model??? How did you determine significance?

L141-146: Describe any formulation used.

>The details of this multivariate analysis can be quite large, so please refer to the cited references.

Table1: Did the author consider to make a balanced division between the compression surface? What I mean by that is. Did the treatments have a common variable (let's say weight) such that radial section and tangential section had the minimal variation possible?

> After checking the wood cutting, I divided it into radial and tangential samples. At that time, I bought a large amount of commercial wood and then selected and sampled. I regretted that I didn't consider the weight. However, there is not much difference between the maximum value, minimum value, and average value, so this time it seems to be good.

OVERALL: There are several important aspects that the authors did not mention.

> Other referees also pointed out. I tried to add new Figure 2 and sentences to make it easier to understand.

RESULTS AND DISCUSSION:

I think the authors did a great job in separating the tables. The authors well presented the results. However, a better discussion needs to be done. Explain the why of things is important for a complete understaing. This was somewhat done from L213 TO L230. But, it not enough.

> If I focused on the loading coefficient obtained from the multivariate analysis, I could deepen my discussion. However, the expression of loading might be confused with the wood load direction and omitted loading coefficient. Furthermore, the number of factors differed depending on the load direction. I was not sure if I could compare them. Therefore, it may not be enough for the reader, but it was considered as much as possible.

OVERALL: It is good, but there is room for improvement.

>I arranged the text a little and improved it.

CONCLUSIONS

We have good points in the midst of the conclusions that can be fully developed as a better sentence. Working more on that.

Good points are: "When considering the application...". "This study indicates..."

> Thank you for your instruction. It will be my encouragement as I'm researching alone while the research environment is not set up. Thank you again.

Round 2

Reviewer 3 Report

This manuscript has improved, but it still needs improvement. In my opinion, it is almost there. 

INTRO:

Even though you have improved the first paragraph, there are elements in the intro that need improvement. Intro is like a funnel, you start broad and goes on narrowing your topic, which NIRS and so on. Now, you repeat the word NIRS multiple times, which makes the reading awkward.

Where is the hypothesis, I get lost. The overall goal? 

Materials and methods

Once again, the section is Materials and Methods, you initiate it with Methods. 

Cryptomeria japonica (L. f.) D. Don?? Check this.

L71-72: Priority was given to the experimental speed, what does this mean?

L75: "was prepared". What kind of preparation?

L76: "artificial drying". Drying is a process of mass transfer. It can be induced artificially, but it not an artificial process. Also, what does industrial in this case mean? Was the temperature set to 102C? 

L75-81: It has improved. However, dealing with wood we need to know more about the material, specially age, where you took the material from. 

When you say "obtained from a commercial Japanese lumber mill", what we need to know is the city so that we have an idea where the material comes from.

L77: Japanese Indutrial standard (JIS) with... 

L79: change to "The total number of analyzed wood samples was 140.

L87: "as described below". When you said that bending test was divided into two types, I would expect that you talked about these types immediately after saying it. If "the bending test method" is a method, include a) or 1)

L88: Notice how many time you write central, centrally, center... 

L97-98: change to "Both Equations are widely used in the wood science field to calculate bending strength." 

L98: ..."Figure 3a displays the setup for the universal machine, where a compression test, ... ... ...". Notice that changing the wording I eliminated the word machine from being repeated. 

L101-102: "Illustrates both the load and force directions..."

Rewrite 103 to 105 sentence. 

L106-107: The DEN was estimated by the division of sample's weight and volume, which was measured by multiplying length and cross-sectional area. 

L112-113: Was this division 70/70?

Figure 4a instead -- Make the changes for all figures, i.e., don't do Figure4(a)

L119: We need to know the air-dry conditions (temperature and relative humidity). 

L124: How or (based on what) did you separated calibration and predictions sets?

Results and discussion

This is the best part of this manuscript. It was well written and designed. I have seen the other reviews. 

Conclusions

It still needs improvement. Please, notice how many times you write bending strength. It makes your writing boring. You could rewrite this section into different paragraphs.

Author Response

#Reviewer 3

This manuscript has improved, but it still needs improvement. In my opinion, it is almost there.

>Thank you for the detailed review, even the trivial phrases in English. I could not do much more about the nuances and expressions of English words. I hope you are satisfied with your point.

INTRO:

Even though you have improved the first paragraph, there are elements in the intro that need improvement. Intro is like a funnel, you start broad and goes on narrowing your topic, which NIRS and so on. Now, you repeat the word NIRS multiple times, which makes the reading awkward.

>Certainly, it looks like items when you point out so. I deleted some sentence (NIRS repeat word).

Where is the hypothesis, I get lost. The overall goal?

>The hypothesis was the detecting the wood anisotropic effect with loading direction by using the NIR acquisition position. I thought the difference of NIR acquisition position was directly affected on the prediction results, but in this study, the difference was less.

The overall goal was the increase with the accuracy of prediction results by NIRS. I have never seen a thesis expression that clearly states a hypothesis.

Materials and methods

Once again, the section is Materials and Methods, you initiate it with Methods.

> I recognized less wood sample information. You only indicated this point. I thought that the description of NIR spectroscopy device was itself “Material” .  It was the difference of interpretation of language.

Cryptomeria japonica (L. f.) D. Don?? Check this.

>Yes, I inserted D.Don.

L71-72: Priority was given to the experimental speed, what does this mean?

>I deleted this sentence.

L75: "was prepared". What kind of preparation?

>I used this sentence as the ready of sample.

L76: "artificial drying". Drying is a process of mass transfer. It can be induced artificially, but it not an artificial process. Also, what does industrial in this case mean? Was the temperature set to 102C?

> It was misleading to write extra information. I deleted some words. I did not have much information of commercial wood sample, I only used general information. Your paper should have the information like the component of mineral cutting saw and the lot of machine.

L75-81: It has improved. However, dealing with wood we need to know more about the material, specially age, where you took the material from.

>I don't know that. I haven't procured wood by specifying it. If I knew it, I'd describe it in more detail.

When you say "obtained from a commercial Japanese lumber mill", what we need to know is the city so that we have an idea where the material comes from.

> It was obscure information. The information of ” located at Fujisawa-city, Kanagawa Pref. “ was added. I don't think where I buy it in Japan, it will have a big change on the result, but in your country, the wood component will change depending on where you buy it.

L77: Japanese Indutrial standard (JIS) with...

>Best. I used Industrial.

L79: change to "The total number of analyzed wood samples was 140.

>Thank you for your wording.

L87: "as described below". When you said that bending test was divided into two types, I would expect that you talked about these types immediately after saying it. If "the bending test method" is a method, include a) or 1)

>I deleted the “as described below”. Thank you for your English usage.

L88: Notice how many time you write central, centrally, center...

> It ’s exactly what you say. I deleted some words. I thought it was easy to understand how many times I wrote.

L97-98: change to "Both Equations are widely used in the wood science field to calculate bending strength."

>Thank you for your suggestion. I used your saying.

L98: ..."Figure 3a displays the setup for the universal machine, where a compression test, ... ... ...". Notice that changing the wording I eliminated the word machine from being repeated.

> Your expression is easy to understand. However, non-natives have a hurdle in telling where and which language they take. Therefore, I expressed it in small parts. I'll be careful.

L101-102: "Illustrates both the load and force directions..."

> Thank you for teaching me.

Rewrite 103 to 105 sentence.

> “The loading surface on the wood was defined as the compression force side. The opposite side was defined as the tensile side. “I used simply expression.

L106-107: The DEN was estimated by the division of sample's weight and volume, which was measured by multiplying length and cross-sectional area.

L112-113: Was this division 70/70?

> Yes, I changed this expression.

Figure 4a instead -- Make the changes for all figures, i.e., don't do Figure4(a)

> I took the parentheses.

L119: We need to know the air-dry conditions (temperature and relative humidity).

> The temperature was ranged 22 - 25 °C, and the relative humidity was 40 - 55 %.

L124: How or (based on what) did you separated calibration and predictions sets?

> The maximum and minimum values of the objective variable were selected into calibration set, and the others were randomly divided into calibration set and prediction set. After division, I calculated the standard deviation of the explanatory variables of each set. If the standard deviation of the objective variable was too different for calibration and prediction, I re-selected the sample. However, there was no clear standard for standard deviation variation in calibration and prediction.

Results and discussion

This is the best part of this manuscript. It was well written and designed. I have seen the other reviews.

> Is that so. It was good.

Conclusions

It still needs improvement. Please, notice how many times you write bending strength. It makes your writing boring. You could rewrite this section into different paragraphs.

> I used six times. I changed “mechanical property” or “physical property”. Once pointed out to you, I thought the representation of physical property in a strict sense was inappropriate. But, I thought the reader understood the expression physical properties in this case.

Best regards.